



# Understanding AMOC stability: the North Atlantic Hosing Model Intercomparison Project

Laura C Jackson[1], Eduardo Alastrué de Asenjo[2,9], Katinka Bellomo[3,4], Gokhan Danabasoglu[5], Helmuth Haak[2], Aixue Hu[5], Johann Jungclaus[2], Warren Lee[6], Virna L Meccia[10], Oleg Saenko[6,8], Andrew Shao[6], and Didier Swingedouw[7]

[1]Met Office, Exeter, UK
[2]Max Planck Institute for Meteorology, Hamburg, Germany
[3]National Research Council of Italy, Institute of Atmospheric Sciences and Climate, Turin, Italy
[4]Polytechnic University of Turin, Department of Environment, Land and Infrastructure Engineering, Turin, Italy
[5]Climate and Global Dynamics Lab, National Center for Atmospheric Research, Boulder, CO 80307, USA
[6]CCCma, Canada
[7]University of Bordeaux, CNRS, Bordeaux, France
[8]SEOS, University of Victoria, BC, Canada
[9]International Max Planck Research School on Earth System Modelling, Hamburg, Germany
[10]National Research Council of Italy, Institute of Atmospheric Sciences and Climate, Bologna, Italy

**Correspondence:** Laura Jackson (laura.jackson@metoffice.gov.uk)

**Abstract.** The Atlantic meridional overturning circulation (AMOC) is an important part of our climate system. The AMOC is predicted to weaken under climate change, however there are theories that it may have a tipping point beyond which recovery is difficult, hence showing quasi-irreversibility (hysteresis). Although hysteresis has been seen in simple models, it has been difficult to demonstrate in comprehensive global climate models. Here we outline a set of experiments designed to explore
AMOC hysteresis and sensitivity to additional freshwater input as part of the North Atlantic hosing model intercomparison project (NAHosMIP). These experiments include adding additional freshwater (hosing) for a fixed length of time to examine the rate and mechanisms or AMOC weakening, and whether the AMOC subsequently recovers once hosing stops.

Initial results are shown from eight climate models participating in the Sixth Coupled Model Intercomparison Project (CMIP6). The AMOC weakens in all models from the freshening, but once the freshening ceases, the AMOC recovers in
half of the models, and in the other half it stays in a weakened state. The difference in model behaviour cannot be explained by the ocean model resolution or type, or by details of subgridscale parameterizations. Nor can it be explained by previously proposed properties of the mean climate state such as the strength of the salinity advection feedback. Instead the AMOC recovery is determined by the climate state reached when hosing stops, with those experiments where the AMOC is weakest not experiencing a recovery.

# 1  Introduction

The Atlantic Meridional Overturning Circulation (AMOC) has an important role in the climate system in transporting heat meridionally. There is evidence from changes in the paleoclimate record, from theories, and from simplified and more complex



models, that the AMOC may be able to experience hysteresis (Clement and Peterson, 2008; McManus et al., 2004; Hawkins et al., 2011; Hofmann and Rahmstorf, 2009; Rahmstorf, 2002, 1996; Rahmstorf et al., 2005). Hysteresis means that a change in

forcing (for example an increase in surface freshwater input) beyond a critical threshold could cause the AMOC to weaken to a weak or reversed state, and that the AMOC would not then recover to its original strength once the forcing was reversed within a human-relevant time frame. If this new state persists for long enough, the model can be considered to have two stable AMOC states (bistability). Although in some models it may be difficult to have a sufficiently long simulation to prove bistability, if the new state persists for centuries it could be said to be quasi-stable.

This behaviour has previously been seen in simplified models, Earth System Models of Intermediate Complexity (EMICs), and a couple of (mostly very low resolution) global climate models (GCMs) (Rahmstorf, 1996; Rahmstorf et al., 2005; Hawkins et al., 2011; Hu et al., 2012), although most previous studies with GCMs found no hysteresis of the AMOC (Stouffer et al., 2006). However, Mecking et al. (2016) and Jackson and Wood (2018) have shown a quasi-stable weak state in a recent GCM with eddy-permitting ocean resolution. Their results showed that if the AMOC in their GCM was weakened sufficiently by

hosing (additional surface freshwater input) in the North Atlantic, that the AMOC then stayed in a weak state for a few hundred years once the hosing was stopped. Understanding the conditions resulting in AMOC hysteresis, and whether they are likely to be met in the future, is very important because a collapse of the AMOC to a weak state would have serious impacts on climate (Jackson et al., 2015).

There are various factors which could contribute to whether a model has a bistable AMOC. Previous results from theories and

simplified models have highlighted the potential importance of the salt advection feedback: if the AMOC exports freshwater (imports salt) from the North Atlantic, then a weakening of the AMOC would reduce this export and freshen the subpolar North Atlantic, resulting in a decrease in buoyancy and a further reduction in AMOC strength (Stommel, 1961; de Vries and Weber, 2005; Drijfhout et al., 2011). Hence if the meridional transport of freshwater by the AMOC (Fov) is negative then the feedback is positive, potentially accelerating the weakening of the AMOC. However, if Fov is positive, then the feedback is

negative, stabilising the strong AMOC state. The importance of the sign of Fov for the AMOC response to freshwater input in GCMs has been shown in several studies (Drijfhout et al., 2011; Jackson, 2013; Liu et al., 2017), however, there are many other feedbacks and processes present in a GCM compared to a simple box model, and these could change, or even remove, the bistability (Jackson, 2013; Weijer et al., 2019). Although Fov is often considered in the South Atlantic (30-34°S), where the waters enter the Atlantic from the Southern and Indian oceans, Mecking et al. (2016) and Jackson and Wood (2018) found

that an important process in the AMOC hysteresis in their model was Fov in the subtropical North Atlantic. Mecking et al. (2016) suggested that this feedback was stronger than in lower resolution models because of higher horizontal ocean resolution leading to a larger magnitude of Fov in the control.

The AMOC response in a GCM may depend on the combination of many different feedbacks (Weijer et al., 2019), and some of these feedbacks may be affected by biases in the model. In particular the value of Fov in the south Atlantic has been shown

to be related to Atlantic salinity biases (Mecking et al., 2017), and both the position of the Atlantic intertropical convergence zone (Liu et al., 2013) and the net Atlantic precipitation (Jackson, 2013) have been shown to affect Fov through changing salinity. Other aspects of the mean state might also affect the AMOC response. For instance, Jackson et al. (2020) showed that





the AMOC response to increasing greenhouse gases could be affected by the location of wintertime deep convection and water mass transformation in the North Atlantic: those models with more deep convection and transformation in the western subpolar gyre in the control were more strongly impacted by warming. Likewise, since sea ice inhibits surface heat loss from the ocean, differences in the location and extent of sea ice could impact the AMOC response.

Results from simplified models and EMICS, have also shown a sensitivity of the AMOC hysteresis to subgridscale parameterizations. Several studies have shown that increased vertical diffusivity can enhance the stability of the AMOC (Schmittner and Weaver, 2001; Prange et al., 2003; Sijp and England, 2006). It has also been shown that when horizontal or along isopycnal diffusion or parameterised eddy-mixing is increased that a greater freshwater input is required to shut down the AMOC (Hofmann and Rahmstorf, 2009; Dijkstra, 2007; Sijp and England, 2009; Sijp et al., 2006).

Many studies have investigated the potential stability of the AMOC through idealised experiments with large freshwater inputs, however, even in a model with a potentially bistable AMOC, the threshold might not be crossed in future scenarios. Hence several studies have investigated more realistic freshwater inputs into the North Atlantic, representing increased melting from glaciers which is not fully included in current GCMs. These show that projections of additional freshwater input from melting glaciers would cause a small additional AMOC weakening before 2100 (Jungclaus et al., 2006; Hu et al., 2009; Bakker et al., 2016; Lenaerts et al., 2015; van den Berk and Drijfhout, 2014). Since this freshwater input occurs around the coasts of Greenland, rather than uniformly, it would primarily freshen the boundary currents. Hence resolving the eddies by using higher resolutions can have an impact on the mixing of the fresh boundary water into the interior where it can impact convection (Weijer et al., 2012; den Toom et al., 2014), and result in a stronger AMOC weakening (Swingedouw et al., 2022).

The North Atlantic Hosing Model Intercomparison Project (NAHosMIP) aims to understand the sensitivity of the AMOC in current GCMs to hosing. Using the experimental setup of Jackson and Wood (2018), in which additional freshwater input is applied to a larger area of the subpolar North Atlantic, we designed a set of experiments to understand whether different GCMs show a similar AMOC hysteresis, with a particular focus on models participating in the current Climate Model Intercomparison Project (CMIP6). Using a large, idealised hosing allows us to understand the model sensitivities and how they differ, and from analysing the similarities and differences between the model responses, we may be able to understand what controls this AMOC response, and how the real world may behave. Although the first set of experiments are very idealised, we also include a set of similar experiments which are less idealised in that they apply the freshwater around the coasts of Greenland only and use a more realistic (though still large) amount of freshwater input. These experiments help us to understand how to apply our understanding of the AMOC sensitivity from the idealised experiments to a less idealised scenario.

Section 2 describes the experimental design of these experiments in detail, with section 3 describing the models which have carried out these experiments so far. Section 4 describes initial results, including how the AMOC responds, how differences in the climate state may influence the AMOC response, and some discussion of what may or may not control the differences between models. A summary of conclusions is given in section 5.



## 2 Experimental Design

Experiments here are based on the preindustrial control experiments (where external forcings are fixed at 1850s conditions) which were conducted as part of the core CMIP6 experiments, described in Eyring et al. (2016). The experiments here use a preindustrial control state as an initial condition and are either hosing experiments (where an additional freshwater flux is applied to the surface ocean) or recovery experiments (with no hosing, but starting from the hosing experiments). A list of experiments is shown in Table 1.

### 2.1 Uniform hosing experiments

To allow a clean comparison with Jackson and Wood (2018), we use the same experimental protocol (uniform hosing from 50°N to the Bering Strait), and use a hosing strength of 0.3 Sv ($1\text{Sv}=10^6\,\text{m}^3/\text{s}$) for our uniform hosing (UH) experiments. This rate of hosing allows us to compare the sensitivity of the different models and the processes and feedbacks involved, and is strong enough that it is more likely that there is a significant response that can be compared.

Tier 1 (highest priority) experiments are a hosing experiment of 0.3 Sv (which should be continued for at least 50 years), and recovery experiments with no hosing spun off from years 20 and 50 of the hosing experiment. These should be run for 100 years unless the AMOC immediately starts recovering, in which case 50 years is sufficient. Tier 2 is to continue the hosing experiment to 100 years with a recovery run after 100 years.

Tier 3 experiments are to repeat the UH experiment and recovery runs for a larger hosing rate of 0.5 Sv. This might be of interest if the AMOC recovers in all the previous experiments.

### 2.2 Greenland hosing experiments

As well as the UH experiments, we also propose a more realistic set of Greenland hosing (GH) experiments where the hosing is applied around Greenland using the method of Gerdes et al. (2006). For this set of experiments a hosing of 0.1 Sv is used, which is considered a large estimate of potential freshwater input from melting glaciers in Greenland (Swingedouw et al., 2007).

Since the surface freshwater will be exported across 50°N and the Bering Strait in different ways due to the different distribution of hosing in the two sets of experiments, using the same magnitude of hosing would not allow a clean comparison. Hence we prefer to pick the rate of hosing most appropriate for each set of experiments.

Tier 1 experiments involve applying a hosing of 0.1 Sv over a region defined around Greenland (see below) for at least 50 years, as well as a recovery experiment with no hosing spun off from year 50 of the hosing experiment. The recovery experiment should again be run for 100 years unless the AMOC immediately starts recovering in which case 50 years is sufficient.

Tier 3 experiments are to continue the hosing experiment to 100 years with a recovery experiment spun off after 100 years.





### 2.3 Hosing fields

#### 2.3.1 Uniform hosing field

To create the hosing field $h$ for the UH experiments we firstly define the region $R1$ which is the region north of $50°$N in the Atlantic to the Bering Strait at $66°$N (including the Arctic).

For a given hosing of $H$ (in m³/s), a hosing field can then be created:

$$h(j,i) = \quad \frac{H}{\int_{R1} dxdy} \quad \text{for j,i} \in R1$$

$$= \quad 0 \quad \text{otherwise}$$

where $dx$, $dy$ are the zonal and meridional grid spacings, and $i$ and $j$ are the zonal and meridional grid coordinates. Hence $\int_{global} h\,dxdy = H$. This hosing field is shown in the top panel of Fig. 1.

#### 2.3.2 Greenland hosing field

Following the protocol defined on http://www.clivar.org/clivar-panels/omdp/core-3 and in Gerdes et al. (2006) we define the
region $R2$ as around the coasts of Greenland from $76°$N on the west side (Nares Strait), down to the tip of Greenland and up to $81°$N on the east side (Fram Strait). See Fig. 3 of Gerdes et al. (2006). The hosing field is then given by

$$h(j,i) = \alpha \exp\left(-r/r_{max}\right) \qquad\qquad \text{for } r \leq r_{max}$$

$$= 0 \qquad\qquad \text{otherwise}$$

where $r$ is the distance perpendicular to the coast, $r_{max} = 300$ km and $\alpha$ is given by

$$\alpha = \frac{H}{\int_{R2} \exp\left(-r/r_{max}\right)dxdy}$$

This hosing field is shown in the bottom panel of Fig. 1.

#### 2.3.3 Applying hosing fields

In the past, most models have used rigid lids or linear free surfaces and have applied freshwater fluxes as virtual salinity fluxes (i.e. they do not add volume to the ocean), however, some models now use a nonlinear free surface which means precipitation adds volume to the ocean. When applying the hosing flux $h$ we do not want to change the volume (which would cause difficulties when applying the compensation), so apply it as a virtual salinity flux $f_s$.

$$f_s(t,j,i) = -\frac{h(j,i)S_0(t,j,i)}{dz_0(j,i)}$$


where $S_0$ is the local salinity in the upper layer, and $dz_0$ is the upper layer thickness. Note the negative sign, since hosing reduces the salinity. Since $h$ has units of m/s, $f_s$ has units PSU/s and is applied to the salinity budget calculation.

$$\frac{dS}{dt}(t, k=0, j, i) += f_s(t, j, i)$$

Note that in the results shown below, CESM2 used a reference salinity rather than the local surface salinity.

### 2.3.4 Compensation

To stop the global salinity from drifting, we apply a volume compensation designed to conserve salt. Firstly we calculate the total flux added:

$$F_{tot}(t) = \int h(j, i) S_0(t, j, i) dx dy$$

and the total ocean volume

$$V = \int_{global} dx dy dz$$

Then the hosing correction applied is

$$\frac{dS}{dt}(t, k, j, i) += F_{tot}(t)/V$$

### 2.4 Other experimental data

We also make use of other experiments conducted as part of CMIP6. The preindustrial control (picon) experiments (documented
in Eyring et al. (2016)) are used as the initial conditions for hosing experiments and used for comparing model mean states. References for data from picon experiments are Danabasoglu et al. (2019); Swart et al. (2019b); Consortium (2019); Ridley et al. (2018, 2019c); Boucher et al. (2018b); Wieners et al. (2019a); Jungclaus et al. (2019a).

The abrupt 4xCO$_2$ experiments quadrupled the CO$_2$ concentrations at the start of the experiment (branching out from the CMIP6 preindustrial controls) and are documented in Eyring et al. (2016). These experiments were conducted by all the models
participating in this study (Swart et al., 2019c; EC-Earth-Consortium, 2019b; Boucher et al., 2018c; Ridley et al., 2019d, 2020; Jungclaus et al., 2019b; Wieners et al., 2019b; Danabasoglu, 2019b), and we also make use of results in other CMIP6 models calculated by Bellomo et al. (2021).

We also use the CMIP6 historical experiments (also documented in Eyring et al. (2016)) to compare AMOC strengths with observed values. For each model an ensemble of historical experiments from 1850-2014 were run, with the ensembles
generated by perturbations of the initial conditions. We calculate the AMOC strength and its uncertainty by using the mean and standard deviation across the available ensemble members. References for the data used are Danabasoglu (2019a); Swart et al. (2019a); EC-Earth-Consortium (2019a); Ridley et al. (2019a, b); Boucher et al. (2018a); Wieners et al. (2019c); Jungclaus et al. (2019c).



## 3 Model description

The experimental protocol described in the previous section has been carried out by several GCMs which have previously
participated in CMIP6. These are all fully coupled climate models. There are 8 GCMs from 6 modelling centres, with two
modelling centres (Met Office and MPI) submitting 2 models with differing resolutions (Table 2). Most use the NEMO ocean
model, but with different atmosphere models, however, there are also GCMs based on the POP (CESM2) and MPIOM (MPI-
ESM-LR and HR) ocean models. Most of the GCMs have a $1°$ horizontal ocean resolution, though HadGEM3-GC3-1MM
has a higher resolution of $1/4°$. Although the two MPI models nominally have very different resolutions, the resolution in the
subpolar North Atlantic is actually similar in both (Jungclaus et al., 2013).

Since previous studies have shown that model representations of eddy mixing and diffusivity can have an impact on AMOC
hysteresis, we include some of these details in Table 3. This includes details of mesoscale and submesoscale parameterizations
and typical values, and parameterizations and typical values for background vertical diffusivity. It should be noted that models
can have a much larger vertical diffusivity in the boundary layer and in certain region from parameterizations of processes such
as shear mixing, convection, double diffusion, tidal mixing.

## 4 Results

### 4.1 AMOC response

#### 4.1.1 Uniform hosing experiments

The AMOC consists of an Atlantic overturning cell with waters in the upper 1000 m travelling northwards on average, sinking
in the North Atlantic, and a southwards return flow between 1000m and 3000m (Fig. 2). Timeseries of the AMOC strength
(defined as the maximum in depth at $26.5°N$) for the set of UH experiments are shown in Fig. 3. During hosing all models
show an AMOC weakening, with a weakening and shallowing of the full AMOC cell (Fig. 2, middle column). In all models
except CanESM5 a strengthening of the deep reversed Antarctic Bottom Water cell is also found.

Although many previous studies have found a relationship between AMOC control strength and AMOC weakening from
increased greenhouse gases (Gregory et al., 2005; Weaver et al., 2007; Winton et al., 2014; Weijer et al., 2020; Bellomo
et al., 2021), this conclusion cannot be made for weakening from hosing. The correlation of the weakening after 100 years of
hosing and the control AMOC strength is not significant (-0.40, p=0.33), although the models with the strongest weakening
are generally those with the strongest control strengths (Fig. 4a). The correlation with percentage AMOC weakening is also
not significant (not shown). We also compare the AMOC weakening from hosing in the u03-hos experiment, with the AMOC
weakening from quadrupling $CO_2$ in the $4xCO_2$ experiment (Fig. 4c). Although there is a significant correlation between the
AMOC weakening in the two experiments (0.73, p=0.04), this high correlation is caused by the two models HadGEM3-GC3-
1MM and CESM2, which have very large weakening in both experiments. It is possible that the processes responsible for the
large AMOC sensitivity in these two models are the same in both scenarios. We note that the AMOC response to increased





$CO_2$ appears to be much stronger in some CMIP6 models in comparison to the previous CMIP5 models (Bellomo et al., 2021),
and that the large weakening in HadGEM3-GC3-1MM and CESM2 are unusual for CMIP5 models, but found in other CMIP6
models. There is also a significant correlation (0.82, p=0.01) between the absolute strength reached after 100 years of hosing
and the absolute strength reached after 150 years of quadrupled $CO_2$ levels (Fig. 4d), which is not caused by outliers. In both
experiments the AMOC initially weakens quickly because of a large change in forcing, followed by more gradual AMOC

weakening as the climate state adjusts. It may be that the feedbacks which oppose AMOC weakening are similar in the two
different scenarios, and hence there is a statistical relationship between the equilibrium AMOC strengths. Data from (Bellomo
et al., 2021) shows that the weak AMOC states here are unusually weak, but within the range of other CMIP5 and CMIP6
models forced with an abrupt quadrupling of $CO_2$. Given that the AMOC weakening compared in Fig. 4c is defined as the
difference between the final AMOC state (compared in Fig. 4d) and the control strength, it is likely that a strong correlation in

one would affect the other comparison. To conclude whether either of both of these conjectures (that the rate of weakening is
similar between the two scenarios, or that the final AMOC state is similar between the two scenarios) is true, would require a
greater number of model results, or improved understanding.

In the recovery experiments where the hosing is stopped after 20 years, the AMOC recovers towards its control state in all
experiments. However, in some experiments the AMOC demonstrates hysteresis by remaining in a weak state for at least 100

years after hosing stops. This occurs for HadGEM3-GC3-1MM and CESM2 after 50 years of hosing, for CanESM5 after 70
years of hosing and for IPSL-CM6A-LR after 100 years of hosing. The streamfunctions of these weak states in the recovery
experiments are shown in Fig. 2 (right column). There are a couple of experiments (IPSL-CM6A-LR and CanESM5 after 50
years of hosing) where the AMOC appears to be staying in a weak state, however, a slight strengthening trend combined with
short timeseries make this conclusion uncertain. We cannot know whether the AMOC weak states are stable, since continuing

the experiments for hundreds to thousands of years is prohibitive, however they remain in the weak state for at least 100 years
so are quasi-stable weak states.

### 4.1.2 Greenland hosing experiments

The second set of experiments performed examine the response of the AMOC to a more realistic freshwater input of 0.1 Sv
around Greenland. Timeseries of AMOC strength are shown in Fig. 5. The AMOC reduction in all experiments is smaller than

that in the previous experiments, likely because the rate of freshwater input is smaller. We note that in additional experiments
with MPI-ESM1-2-LR which use 0.3 Sv of hosing around Greenland, the AMOC weakening is slightly smaller than when
applying the same amount uniformly.

The location of the freshwater input around Greenland, rather than spread uniformly over the subpolar Atlantic and Arctic,
could potentially be important, since the fresh anomalies can be easily exported from the subpolar North Atlantic by boundary

currents, and require resolved or parameterised eddies to mix the freshwater into the interior where it impacts deep convection
(Weijer et al., 2012; Swingedouw et al., 2022). Recent research has suggested that hosing around Greenland might be more
effective in ocean models with a higher horizontal resolution, since the eddy mixing of the freshwater from boundary currents
around Greenland into the interior might be stronger, resulting in stronger inhibition of deep convection (Swingedouw et al.,





2022). Although HadGEM3-GC3-1MM (which has the highest horizontal ocean resolution) has a strong AMOC weakening,
there is a stronger AMOC weakening in CESM2, which has a lower resolution (Table 2).

A comparison of the AMOC weakening from both UH and GH experiments shows that those models with a stronger weak-
ening from one hosing scenario have a stronger weakening from the other hosing scenario (Fig. 4b). Although the correlation is
significant (0.90, p<0.01) there is a large difference in AMOC weakening in the g01-hos experiment in CESM2 and HadGEM3-
GC3-1MM, despite similar weakening in the u03-hos experiment. This suggests that the geographical distribution of hosing
might impact different models differently.

## 4.2  Exploration of the AMOC threshold

The experiments where the AMOC does not recover are the experiments where the AMOC has reached the weakest values (see
Fig. 6 and 7). Although there is a clear separation at 26.5°N between the experiments where the AMOC recovers and where
it does not when considering the first decade after hosing stops (Fig. 6), there is some overlap when considering the AMOC
strength in the decade before the hosing stops (Fig. 7). Decadal means are used here to limit the impact of internal variability.
There is a clearer separation when considering the AMOC at 45°N, with those experiments where the AMOC weakens to less
than 5 Sv not showing a recovery (though for CanESM5 after 50 years of hosing, the results are uncertain). We note that in
the previous ensemble of HadGEM3-GC2 experiments, a threshold of 8 Sv (for AMOC at 26.5°N) was found (Jackson and
Wood, 2018). The clear separation seen between the experiments where the AMOC recovers and those where it does not, is
not found if we instead use AMOC anomalies or fractional changes, rather than absolute values. Although the absolute values
that the AMOC reaches during hosing appear to determine whether the AMOC subsequently recovers or remains in a weak
state, it should be noted that in experiments with HadGEM3-GC2 where the AMOC is weakened to 4 Sv by increasing $CO_2$,
the AMOC subsequently recovers when $CO_2$ increases are reversed (not shown). Hence it is likely that the weak AMOC state
is indicative of other changes to the climate during hosing which sustain the weak state, but that these changes differ under
weakening from increased $CO_2$.

For those experiments where the AMOC does not recover, they also tend to have the weakest March mixed layer depths
(MLD) and the smallest March mixed volume (measured as the volume of water between the MLD and 100m), both of which
are measures of deep convection (Fig 7), in particular the MLD in the Greenland-Irminger-Norwegian (GIN) seas appears to
be indicative of whether the AMOC will subsequently recover when hosing stops.

If we consider all the states that initialise the recovery experiments (taken as decadal means before the hosing stops), we
can group these into states where the AMOC subsequently recovers when hosing stops (named $\mathbb{S}_R$), and those states where
the AMOC stays weak (named $\mathbb{S}_W$). These states are characterised by the model and the number of years of hosing (using
the UH 0.3 Sv hosing scenario) and are shown in Table 4. Examination of these states reveals differences in the sea surface
temperatures (SST) and salinities (SSS) between these two groups. The annual mean SST for all the states in $\mathbb{S}_R$ and $\mathbb{S}_W$ (see
Table 4) are shown in the top panels of Fig. 8. Those states where the AMOC does not recover are, on average, colder (and
fresher, not shown) than those states where the AMOC does recover. The middle panels of Fig. 8 then show the coldest SST
for each grid point of the states in $\mathbb{S}_R$, and the warmest SST of the states in $\mathbb{S}_W$. The bottom panels show the difference in





the states in the decade before hosing stops: in the north east Atlantic and eastern Greenland-Irminger-Norwegian (GIN) seas, all states where the AMOC subsequently recovers are warmer and saltier (and denser, not shown) than all the states where the
AMOC stays weak. In the states the AMOC recovers from, there is also generally less winter sea ice in the north east Atlantic (see contours, middle panels of 8).

### 4.3   Are there intrinsic model differences affecting the AMOC threshold?

One fundamental question is why the AMOC recovers after hosing in some models but not in others. We note that we find AMOC hysteresis in half the CMIP6 models tested (Table 4) and that this is not dependent on the ocean model used or the
horizontal ocean resolution (Table 2). Since several studies have shown that increased vertical diffusivity can make the AMOC more stable (Schmittner and Weaver, 2001; Prange et al., 2003; Sijp and England, 2006), we examine the parameterizations and values of background vertical diffusivity in Table 3. There is no clear relationship to the models showing AMOC hysteresis, however we note that GCMs have other parameterizations which enhance the vertical diffusivity near the surface and in particular locations (i.e. convection) which are not captured by these background values. Studies have also shown that increased
mesoscale eddy advection and horizontal or isopycnal diffusion can also stabilise the AMOC (Hofmann and Rahmstorf, 2009; Dijkstra, 2007; Sijp and England, 2009; Sijp et al., 2006), however Table 3 also shows no obvious relationship of the typical values or parameterization scheme of mesoscale advection or diffusion with those models where the AMOC shows hysteresis.

Previous studies have suggested that the freshwater transport by the AMOC (Fov) in the South Atlantic is important for AMOC stability, since if the AMOC exports freshwater, then weakening the AMOC would result in less freshwater exported,
and hence a freshening and further weakening of the AMOC (Rahmstorf, 1996; Drijfhout et al., 2011). Mecking et al. (2016) argued that the greater export of freshwater in their model (HadGEM3-GC2, a forerunner of HadGEM3-GC3-1MM) at 30°N was responsible for the AMOC not recovering after hosing, and suggested that increased horizontal ocean resolution in their model increased Fov in the North Atlantic subtropics and strengthened this feedback. However, in the experiments in this study it can be seen that there are models with lower horizontal resolution (1° as opposed to 0.25°) where the AMOC also does not
recover, so this is not a feature of higher resolution ocean models only. We also do not find any systematic difference in the control values of Fov at any latitude between those models where the AMOC recovers and those where it does not (Fig. 9), suggesting that this advective feedback does not determine which models show hysteresis.

Jackson et al. (2020) found that the AMOC response to increased greenhouse gases was sensitive to the amount of water mass transformation (WMT) and mixed layer depth (MLD) in present day controls in the west subpolar North Atlantic in a
set of climate models (see also Koenigk et al. (2021)). Comparisons here of March MLD in the control experiments reveal no systematic difference between the models where the AMOC recovers and those where it remains weak (Fig. 10). Also a comparison of WMT in some of the models considered here (Jackson and Petit, 2022) shows no systematic differences between models, with CanESM5 having little WMT in the Labrador and Iceland-Irminger seas and HadGEM3-GC3-1MM having strong WMT, compared with other models. Likewise, there is no systematic difference between the two sets of models
in terms of control AMOC strength (Fig. 3), or control SST, SSS or ice extent (not shown).





Another model bias that might affect the AMOC response to hosing is the boundary between the subpolar and subtropical gyres. Swingedouw et al. (2013) proposed that in models where the boundary is more zonal, freshwater that is added to the subpolar gyre is more easily exported to the subtropical gyre via the Canary current in the eastern subtropical Atlantic. Inspection of the barotropic streamfunction for the models in this study reveals no systematic differences in the strengths of

the subpolar or subtropical gyres, or in the gradient of the intergyre boundary between the two sets of models. This might be due to the fact that uniform hosing has been used here, while Swingedouw et al. (2013) were analysing freshwater put around Greenland and the way it spread in the North Atlantic.

Given that whether the AMOC recovers after hosing is dependent on the strength it reaches during hosing, the AMOC recovery could be dependent on both the AMOC strength in the control and its sensitivity to hosing (the amount of weakening

it experiences). We note that IPSL-CM6A-LR and CanESM5 start from a state with a weak AMOC (11-12 Sv) and experience a small weakening of 5-6 Sv, whereas HadGEM3-GC3-1MM and CESM2 have a relatively strong AMOC in the control (16-18 Sv) and experience a large weakening (10-11 Sv). Present day observations of the AMOC (Frajka-Williams et al., 2021) suggest a value of 16.8 Sv over the period 2005-2014. The strength of the AMOC in these models over this period is shown in Table 2. While the observational value lies within model uncertainty for most models, CanESM5 and IPSL-CM6A-LR

have very weak AMOC strengths. Hence of those models with a reasonable AMOC strength, only those with a strong AMOC weakening from hosing reach a state where the AMOC does not recover.

## 5 Conclusions

We have presented the experimental protocol for the NAHosMIP project, which aims to understand the sensitivity of the AMOC to additional freshwater in the North Atlantic. We show that about half the CMIP6 models which run this protocol find

states where the AMOC does not recover after hosing of 0.3 Sv. The difference in model behaviour cannot be explained by the ocean model resolution or type, by details of subgridscale parameterizations, or by aspects of the mean climate state such as the strength of the salinity advection feedback, location or depth of deep convection, or the position of the intergyre boundary.

Instead, the AMOC behaviour appears to be related to the state the model reaches after hosing finishes: those experiments where the AMOC has reached the weakest states, where March mixed layer depths are the shallowest, and where the eastern

subpolar gyre and Nordic seas are the coldest and freshest with greatest sea ice extent, are those where the AMOC subsequently does not recover.

Given that AMOC strength after hosing is a good indicator of AMOC recovery, it may be possible to relate AMOC recovery to the combination of AMOC control strength, sensitivity of the AMOC to freshwater forcing, and the duration of the freshwater input.

These results are all from experiments which apply an unrealistically large, idealised freshwater input. So, although they are useful for understanding the model sensitivity, they should not be regarded as future scenarios. Results are also shown from an additional set of experiments, where a large (but not unrealistic) freshwater flux is applied around Greenland to simulate freshwater from melting ice sheets. These results show AMOC weakening which varies a lot across models, with some models

showing no weakening and others showing a weakening of several Sv in 50 years. While the AMOC shows no hysteresis from
this less idealised forcing, a sustained weakening would still have large impacts on climate.

Future studies will be examining the mechanisms involved in the AMOC recovery, to improve our understanding of the
important feedbacks involved, as well as studies to examine the impacts of a sustained AMOC weakening. We also hope to
use this protocol for future experiments with higher resolution climate models, which improve the resolution of eddies and
boundary currents in the subpolar North Atlantic. Understanding how the models' responses to freshwater forcing evolve in
the presence of warming is also a future research direction.

*Code and data availability.* Code and data used to do the analysis (including annual mean AMOC streamfunction for all models) and
code to plot figures in this manuscript are available from https://doi.org/10.5281/zenodo.7324394. Other data from preindustrial con-
trol experiments is available via the Earth System Grid Federation (ESGF) servers with information on obtaining data available from
https://pcmdi.llnl.gov/CMIP6/Guide/dataUsers.html. Code for creating hosing files and compensation, and sample files are available from
https://doi.org/10.5281/zenodo.7225014

*Author contributions.*

LJ led the paper, did the analysis and wrote the manuscript. All authors conducted the experiments, provided the diagnostics
and edited the manuscript.

*Competing interests.*

The authors declare that there are no competing interests.

*Acknowledgements.* LJ was supported by the Met Office Hadley Centre Climate Programme funded by BEIS. KB has received funding
from the European Union's Horizon 2020 research and innovation programme under the Marie Skłodowska-Curie grant agreement No.
101026907 (CliMOC). AH was supported by the Regional and Global Model Analysis (RGMA) component of the Earth and Environmental
System Modeling Program of the U.S. Department of Energy's Office of Biological & Environmental Research (BER) via National Science
Foundation IA 1844590. The EC-Earth3 simulations shown in this work were carried out at ECMWF under the special projects SPITBELL
and SPITMEC2. This material is based upon work supported by the National Center for Atmospheric Research (NCAR), which is a major
facility sponsored by the US National Science Foundation under Cooperative Agreement 1852977. This project is TiPES contribution number



166: this project has received funding from the European Unions Horizon 2020 research and innovation programme under grant agreement

No 820970.



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





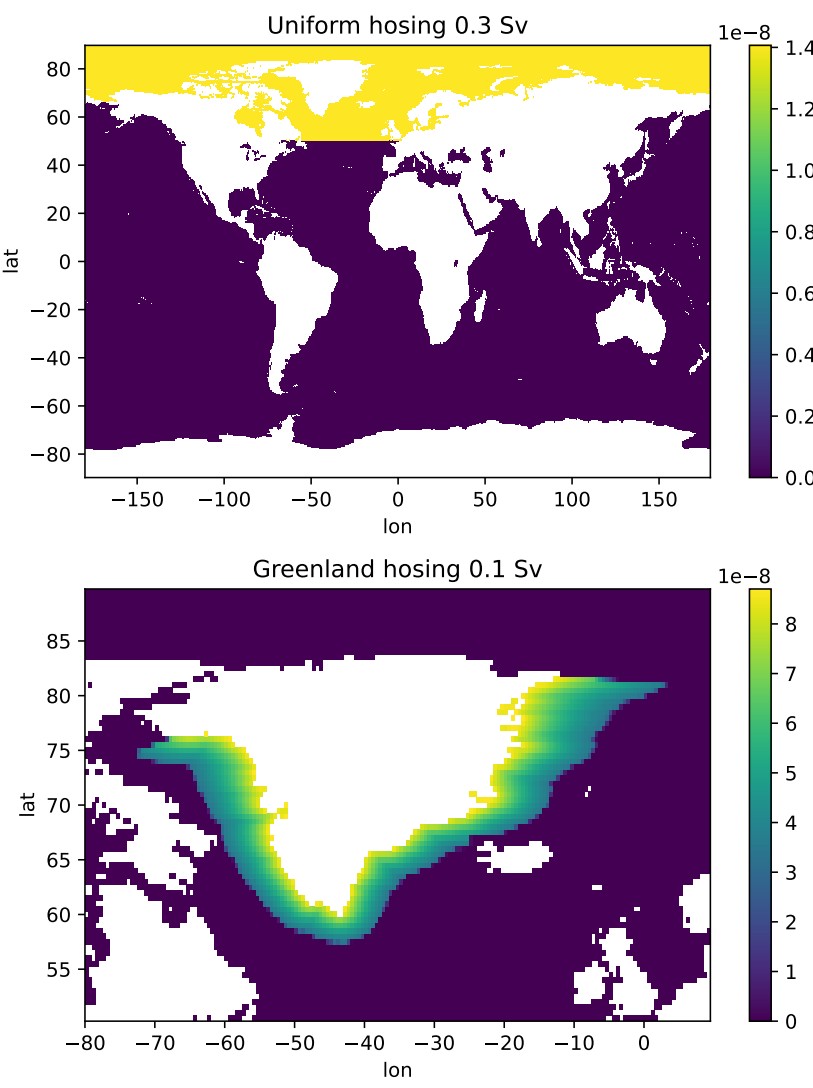

**Figure 1.** Hosing fields. Top panel shows the hosing field used for u03-hos. Bottom panel shows the hosing field used for g01-hos. Units are m/s.





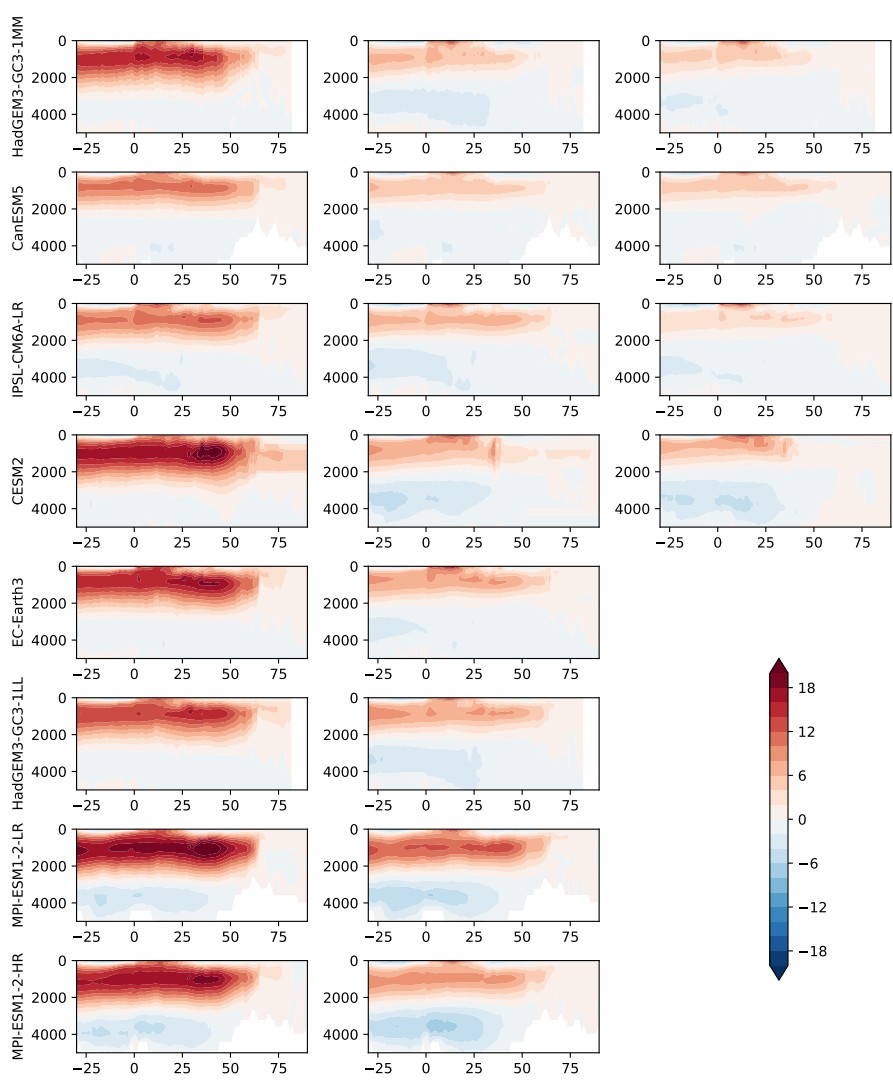

**Figure 2.** Time mean streamfunctions (Sv). Each row shows AMOC streamfunctions from different models. The first column shows the control, the second column years 40-50 of u03-hos, and the third column shows year 50-100 of the recovery experiments where the AMOC stays weak. For HadGEM3-GC3-1MM and CESM2 this is of experiment u03-r50, for IPSL-CM6A-LR this is of experiment u03-r100 and for CanESM5 this is of experiment u03-r70.





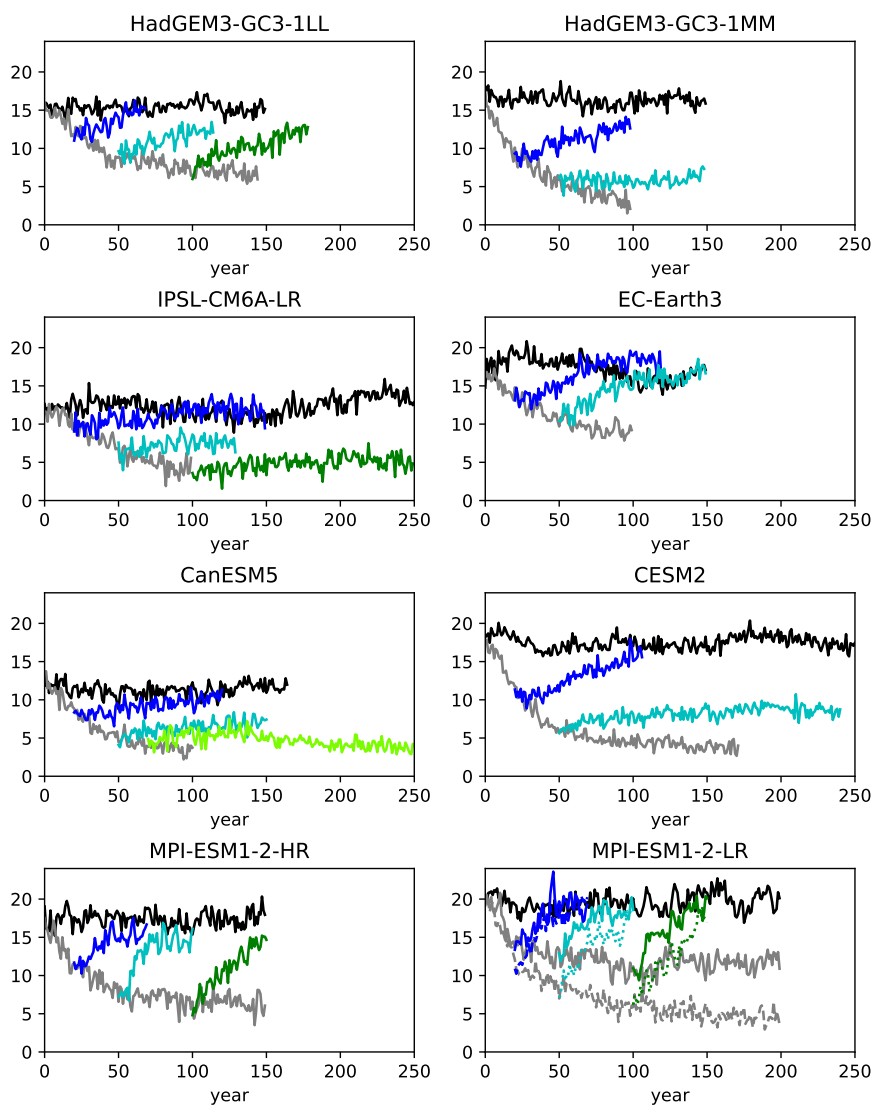

**Figure 3.** AMOC strength (maximum in depth at 26.5°N) for UH experiments. Each panel shows experiments conducted with different models. Experiments are the control (black), u03-hos (grey), u03-r20 (blue), u03-r50 (cyan), u03-r70 (light green) and u03-r100 (green). MPI-ESM1-2-LR also shows the same experiment with a stronger hosing rate of 0.5 Sv (dashed lines).





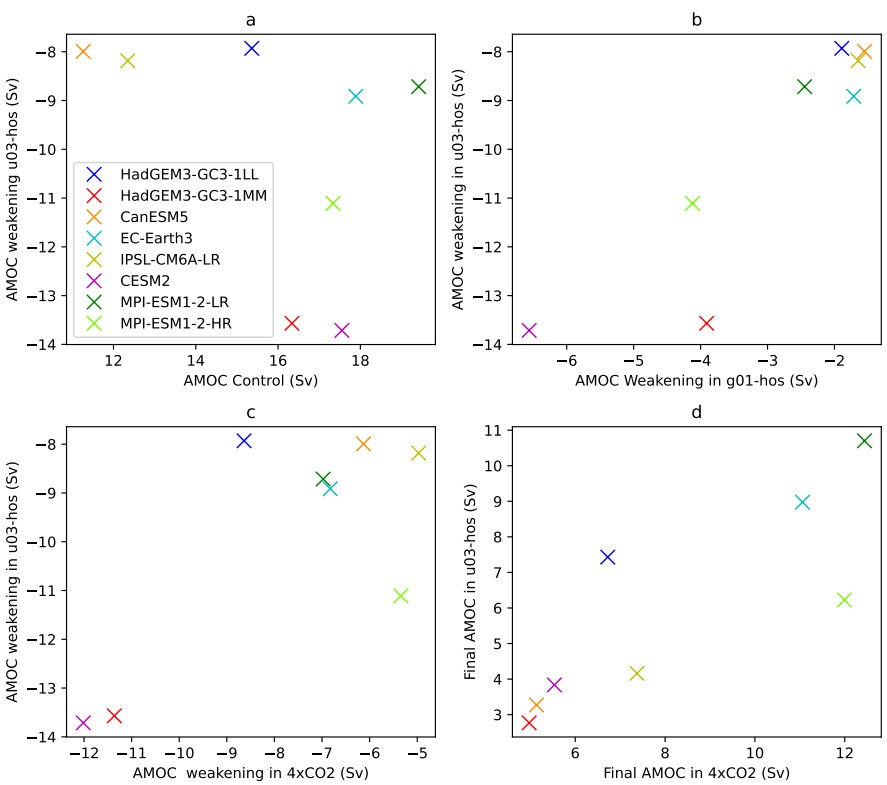

**Figure 4.** Relationships of AMOC weakening. a) Comparison of AMOC anomaly (year 90-100 mean in u03-hos) with the AMOC strength in the control. b) Comparison of AMOC anomalies in u03-hos (year 90-100) and in g01-hos (year 40-50). c) Comparison of AMOC anomalies (year 90-100) in u03-hos and (year 140-150) in the 4xCO2 experiment. d) As c but showing final AMOC strengths rather than anomalies. Colors indicate different models.

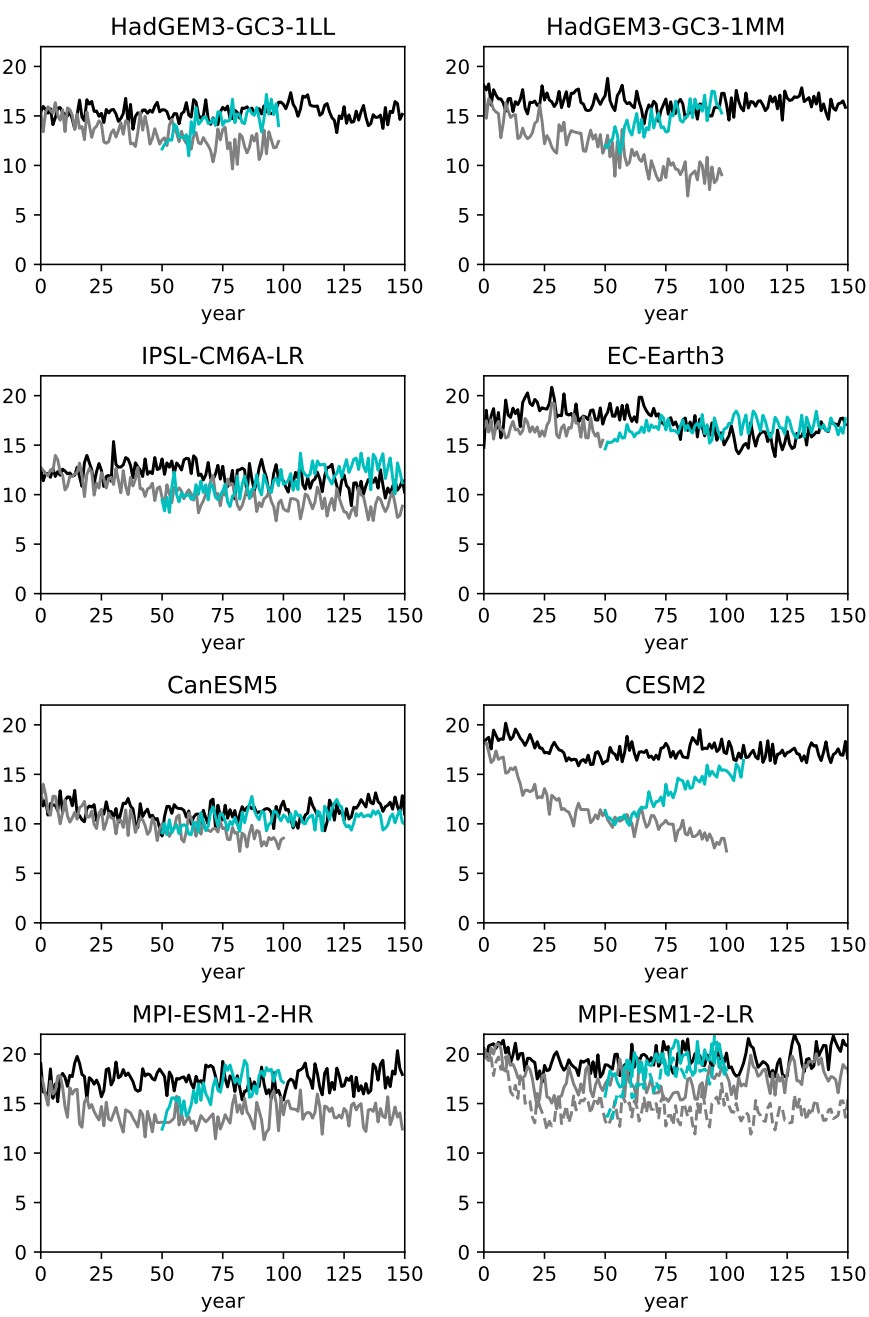

**Figure 5.** AMOC strength (maximum in depth at 26.5°N) for GH experiments. Each panel shows experiments conducted with different models. Experiments are the control (black), g01-hos (grey) and g01-r50 (cyan). MPI-ESM1-2-LR also shows the same experiment with a stronger hosing rate of 0.3 Sv (dashed lines).



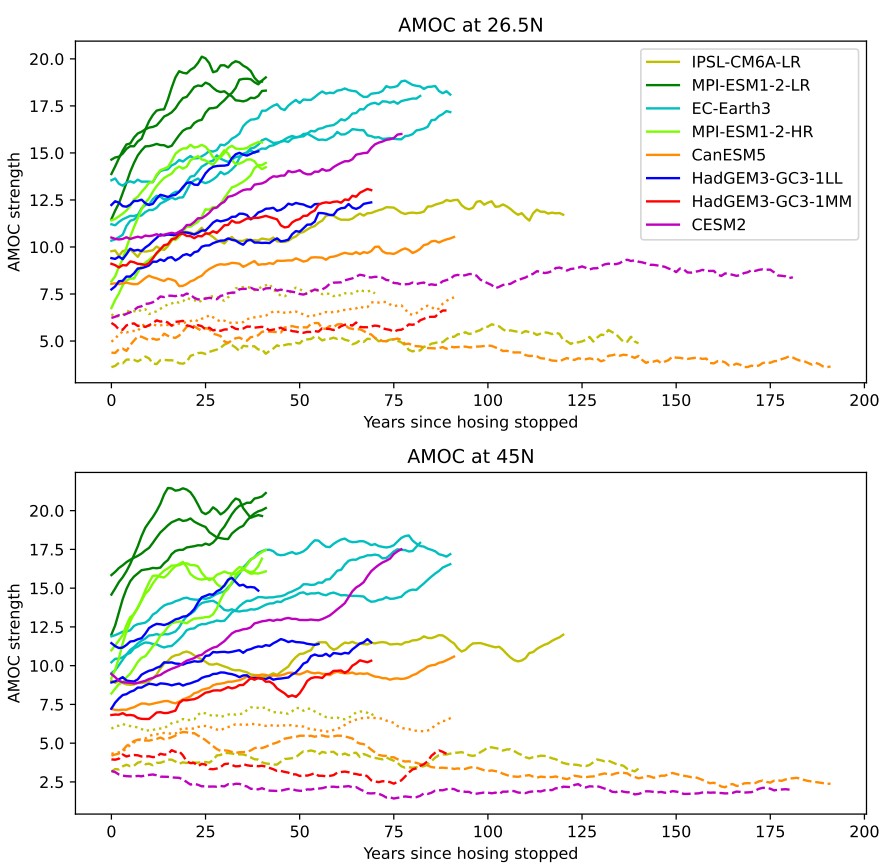

**Figure 6.** AMOC timeseries in recovery experiments plotted as ten year running means. All recovery experiments in the set having experienced UH of 0.3 Sv, with plotted timeseries starting from the time hosing stops. Colors show the model, with solid lines indicating experiments where the AMOC recovers, dashed lines indicating experiments where the AMOC stays in a weak state, and dotted lines indicating experiments where the AMOC response is uncertain. Top panel shows the AMOC measured at 26.5°N and the bottom panel shows the AMOC at 45°N (maximum in depth).



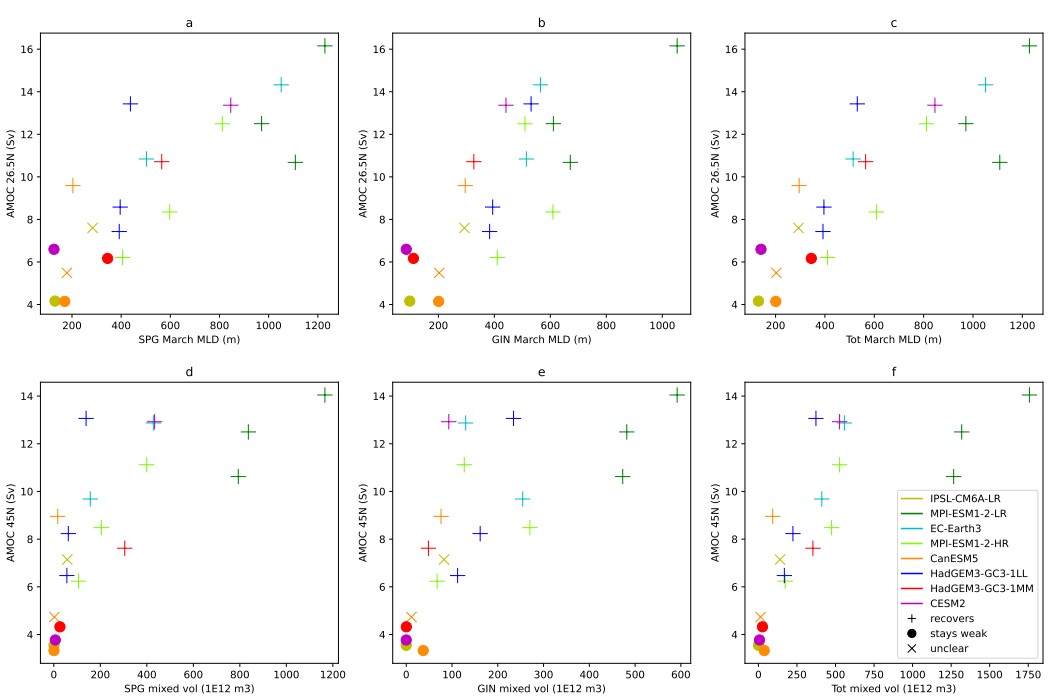

**Figure 7.** Scatter plots of decadal means from the decade before hosing stops. Top panels: AMOC at 26.5°N against maximum mean March MLD over a) the subpolar gyre (80°W-20°E, 50-65°N), b) the GIN seas (50°W-40°E, 65-80°N), c) the whole region (80°W-40°E, 50-80°N). Bottom panels: As top panels but for the mixed volume defined as the volume above the mixed layer depth and below 100m.





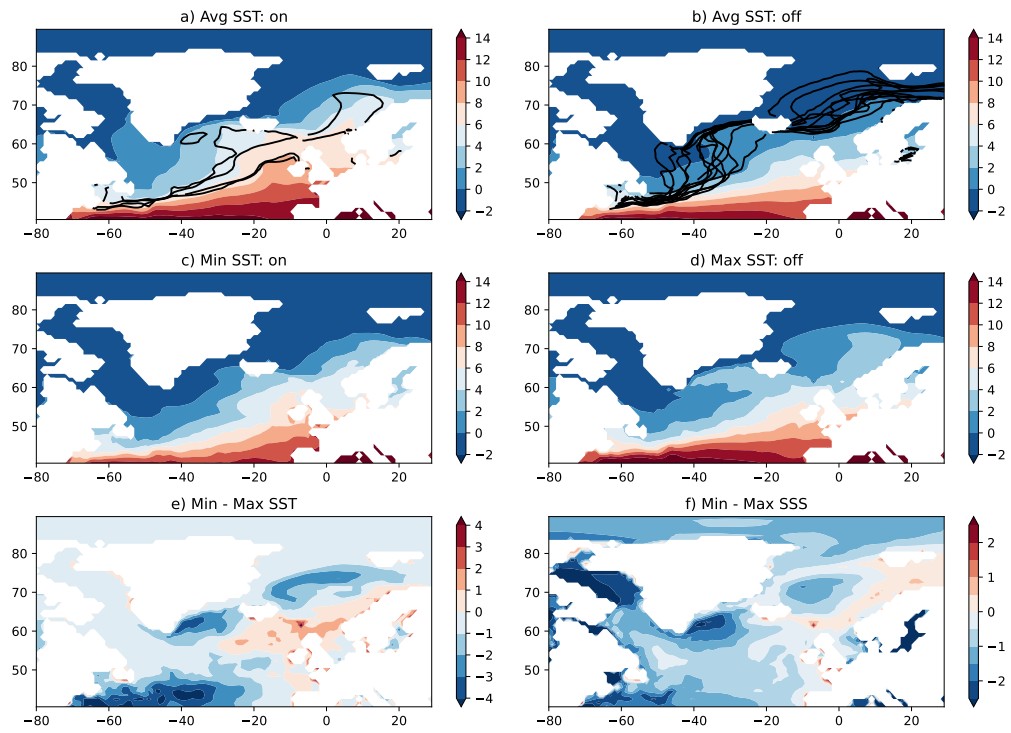

**Figure 8.** Top panels show the mean of SST (°C) of (a) all the states in $\mathbb{S}_R$ and (b) states in $\mathbb{S}_W$ (see Table 4). Black lines show winter sea ice extent (where concentration > 20%) from these states. Middle panels show (c) the minimum SST at each grid point for states in $\mathbb{S}_R$ and (d) maximum SST at each grid point for states in $\mathbb{S}_W$. Panel e shows the difference c)-d). Panel f shows the equivalent to panel e) but for SSS (PSU).





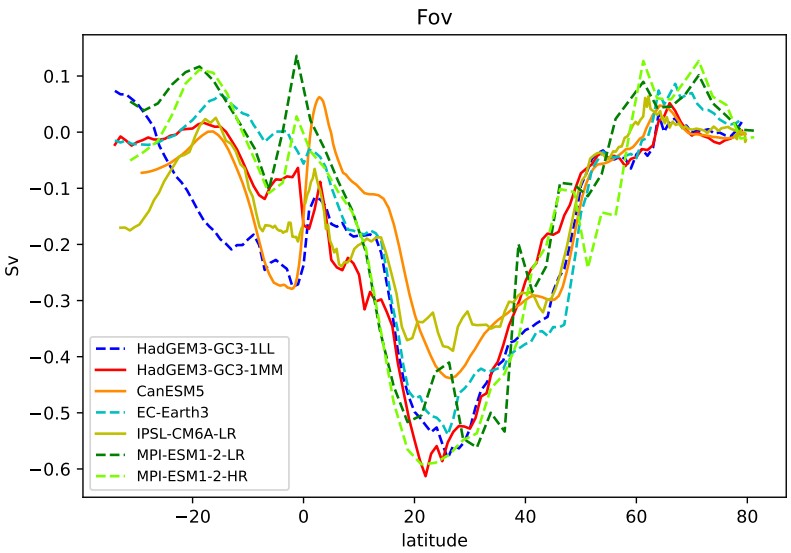

**Figure 9.** Values of Fov (the freshwater transport by the AMOC in the Atlantic, Sv) by latitude for the control experiments. Different colours indicate different models, and solid/dashed lines indicate those models which have/do not have experiments where the AMOC stays weak after hosing stops.

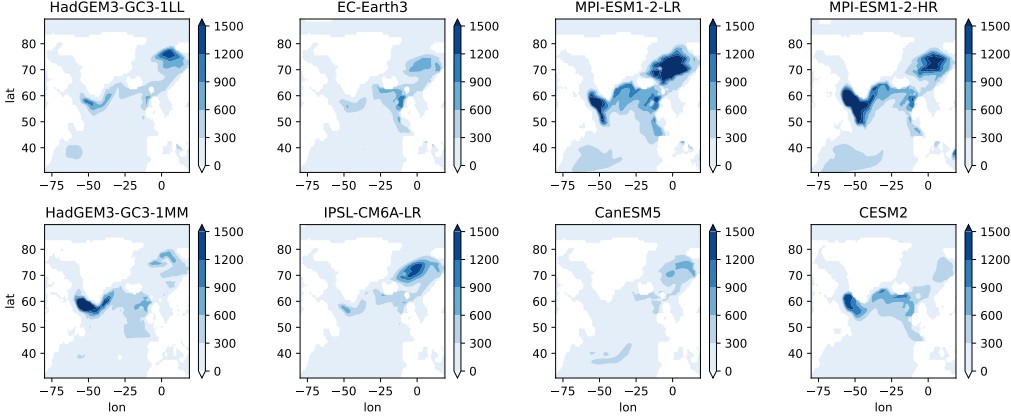

**Figure 10.** Time-mean March mixed layer depth (m) in the control experiments. Models in the upper panels are those where the AMOC always recovers during subsequent recovery experiments, and models in the bottom panels are those where the AMOC stays in a weak state in at least one experiment.

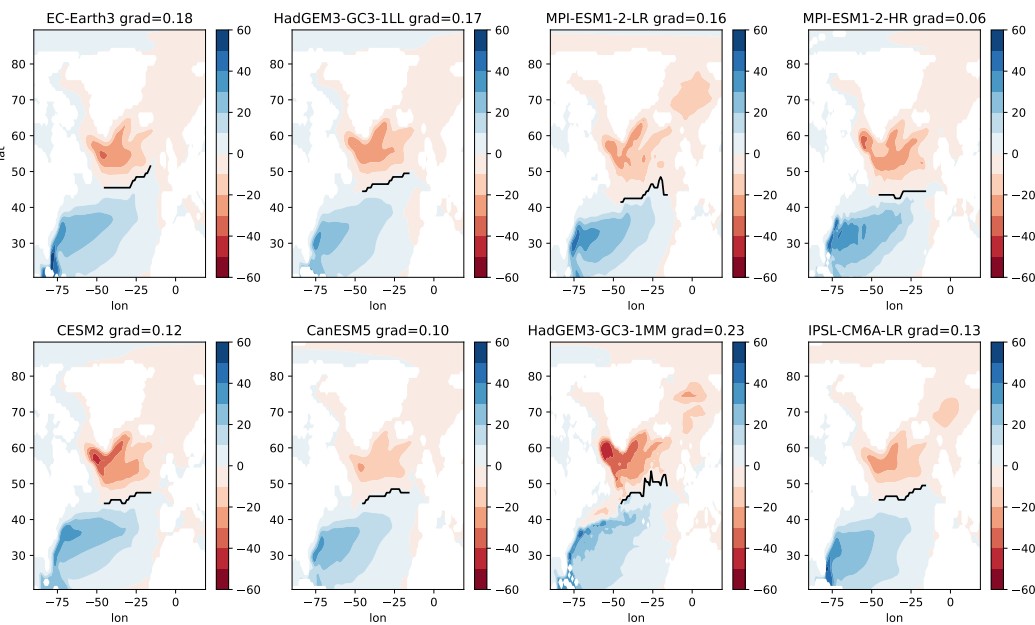

**Figure 11.** Time-mean barotropic streamfunction (Sv) in the control experiments. Models in the upper panels are those where the AMOC always recovers during subsequent recovery experiments, and models in the bottom panels are those where the AMOC stays in a weak state in at least one experiment. Black lines show the intergyre boundary between the subtropical and subpolar gyres over 15-45°W, with the gradient of the boundary listed in the panel titles (degrees latitude per degree longitude).





**Table 1.** Experiments

| Experiment set | Experiment name | Description | Tier |
|---|---|---|---|
| Control | picon | CMIP6 preindustrial control | 1 |
| 0.3Sv UH | u03-hos | uniform hosing of 0.3 Sv | 1 |
| | u03-r20 | no hosing spun off from year 20 of u03-hos | 1 |
| | u03-r50 | no hosing spun off from year 50 of u03-hos | 1 |
| | u03-r70 | no hosing spun off from year 70 of u03-hos | 2 |
| | u03-r100 | no hosing spun off from year 100 of u03-hos | 2 |
| 0.5 Sv UH | u05-hos | uniform hosing of 0.5 Sv | 3 |
| | u05-r20 | no hosing spun off f5om year 20 of u05-hos | 3 |
| | u05-r50 | no hosing spun off from year 50 of u05-hos | 3 |
| | u05-r100 | no hosing spun off from year 100 of u05-hos | 3 |
| 0.1 Sv GH | g01-hos | Greenland hosing of 0.1 Sv | 1 |
| | g01-r50 | no hosing spun off from year 50 of g01-hos | 1 |
| | g01-r100 | no hosing spun off from year 100 of g01-hos | 3 |

**Table 2.** Models used in this study. First column lists the models. Second column shows the AMOC strength in the historical experiments over year 2005-2014, with the mean and twice the standard deviation of the ensemble values. Subsequent columns list the component atmosphere, ocean and sea ice models, the nominal ocean resolution number of vertical levels, and references. Those models which have demonstrated an AMOC hysteresis in this study are in bold.

| Model | AMOC (Sv) | Atm model | Ocean model | Ice model | Ocean resolution, levels | Reference |
|---|---|---|---|---|---|---|
| HadGEM3-GC3-1LL | $16.4 \pm 1.6$ | UM GA7 | NEMO3.6 | CICE GSI8.1 | $1°,75$ | Williams et al. (2018) |
| **HadGEM3-GC3-1MM** | $16.0 \pm 0.8$ | UM GA7 | NEMO3.6 | CICE GSI8.1 | $0.25°,75$ | Williams et al. (2018) |
| **CanESM5** | $12.3 \pm 0.6$ | CanAM5 | NEMO3.4.1 | LIM2 | $1°,45$ | Swart et al. (2019d) |
| EC-Earth3 | $17.3 \pm 2.0$ | IFS 36r4 | NEMO3.6 | LIM3 | $1°,75$ | Döscher et al. (2022) |
| **CESM2** | $18.3 \pm 0.6$ | CAM6 | POP2 | CICE5 | $1°,60$ | Danabasoglu et al. (2020) |
| **IPSL-CM6A-LR** | $12.4 \pm 1.2$ | LMDZ6 | NEMO3.6 | LIM3 | $1°,75$ | Boucher et al. (2020) |
| MPI-ESM-LR | $17.8 \pm 1.2$ | ECHAM6.3 | MPIOM | MPIOM | $1.5°,40$ | Mauritsen et al. (2019) |
| MPI-ESM-HR | $16.8 \pm 1.2$ | ECHAM6.3 | MPIOM | MPIOM | $0.4°,40$ | Müller et al. (2018) |



**Table 3.** Parameterizations of mixing with typical values in m$^2$/s. For mesoscale advection and diffusion models use either the formulations in Gent and McWilliams (1990, GM90) and Redi (1982, R82) or the formulation in Griffies (1998, G98). Those models with a submesoscale parameterization use Fox-Kemper et al. (2011). For vertical diffusivity models either use the turbulent kinetic energy scheme described in Madec and Team (2012, TKE12) or or Madec and Team (2016, TKE16) , or use the schemes in Pacanowski and Philander (1981, PP) or Large et al. (1994, KPP). Those models which have demonstrated an AMOC hysteresis in this study are in bold.

| Model | Mesoscale adv | Mesoscale diff | Submesoscale | Background vert diff |
|---|---|---|---|---|
| HadGEM3-GC3-1LL | GM90, $\leq$1000 | R82, 1000 | No | TKE, $1.2 \times 10^{-6} - 1.2 \times 10^{-5}$ |
| **HadGEM3-GC3-1MM** | No | R82, 150 | No | TKE, $1.2 \times 10^{-6} - 1.2 \times 10^{-5}$ |
| **CanESM5** | GM90, 100-2000 | R82, $\leq$1000 | No | TKE, $5 \times 10^{-6}$ |
| EC-Earth3 | GM90, $\leq$1000 | R82, $\leq$1000 | Yes | TKE, $1 \times 10^{-6} - 1.2 \times 10^{-5}$ |
| **CESM2** | G98, 300-3000 | G98, 300-3000 | Yes | KPP, $1 \times 10^{-6} - 3 \times 10^{-5}$ |
| **IPSL-CM6A-LR** | GM90, $\leq$1000 | G98, $\leq$1000 | Yes | TKE, $1 \times 10^{-6} - 1.2 \times 10^{-5}$ |
| MPI-ESM-LR | G98, $\leq$250 | G98, $\leq$1000 | No | PP, $1.1 \times 10^{-5}$ |
| MPI-ESM-HR | G98, $\leq$250 | G98, $\leq$250 | No | PP, $1.1 \times 10^{-5}$ |

**Table 4.** Length of hosing which determines whether the AMOC stays weak or recovers after 0.3 Sv UH. States where the AMOC subsequently recovers when hosing stops ($\mathbb{S}_R$) are listed in the middle column, and those states where the AMOC stays weak ($\mathbb{S}_W$) are listed in the right column.

| Model | Length of hosing for AMOC recovery | Length of hosing for AMOC staying weak |
|---|---|---|
| CanESM5 | 20 | 70 |
| CESM2 | 20 | 50 |
| EC-Earth3 | 20,50 | |
| HadGEM3-GC3-1MM | 20 | 50 |
| HadGEM3-GC3-1LL | 20,50,100 | |
| IPSL-CM6A-LR | 20 | 100 |
| MPI-ESM1-2-HR | 20,50,100 | |
| MPI-ESM1-2-LR | 20,50,100 | |