# Peer review of "Understanding AMOC stability: the North Atlantic Hosing Model Intercomparison Project"

_Geoscientific Model Development, 2022_

## Author Response (AR1)

*We thank the reviewers for their comments on our manuscript and address their points below.*

Reviewer 1.

Review of: Understanding AMOC stability: the North Atlantic Hosing Model Intercomparison Project, by Jackson et al.

In this paper the authors introduce a protocol for systematically investigating the stability of the Atlantic Meridional Overturning Circulation in climate models. This protocol includes two distinct experimental procedures, where anomalous freshwater is either applied over the entire Arctic, or around Greenland. The authors present initial results from 8 coupled climate models that participated in this NAHosMIP experiment. They compare the AMOC response to these freshwater prescriptions, and try to identify an indicator to predict whether the AMOC will recover or not.

The paper is short and to the point and is well written. It succeeds in its main goal of introducing the NAHosMIP protocol, and this paper forms a solid basis for more detailed analysis. I recommend accepting this paper after some minor revisions.

General comment:

- Just a general comment: The paper does not make the claim that the weakened AMOC state is a collapsed state, and that seems to be the right approach. The weakened AMOC structures in Fig. 2 (middle column) just seem weakened versions of the control AMOC states (left column), and do not seem topologically different. In other words, they don't look like a collapsed or reversed circulation as one would expect from a true off-state --as for instance in simple box models, or in the bifurcation diagrams from the Dijkstra group (e.g., Fig. 2 of Huisman et al. 2010). Instead, they seem to be more representative of the 'cold on' state of the glacial period (Fig. 8 in Weijer et al. 2019) in which any convection was pushed southward. This may be a useful distinction to make.

*We agree that there seem to be similarities with the cold on state, however since we haven't done more detailed comparisons we do not discuss this in the manuscript. However, we have added a brief discussion that the weak state does not look like a reversed off state (end of 4.1.1)*

Specific comments:

- p. 1, l. 2: "...there are theories..." -> "...theories suggest..."?

*Done*

- p. 1, l. 7: or -> of

*Done*

- >p. 2, l. 26: "…a couple of…" -> "…several…"?

*Done*

- p. 2, ll. 34-47: "… North Atlantic…": Obviously the salt advection in the North Atlantic is a strong positive feedback, but as far as I am aware, only Fov on the southern boundary of the Atlantic (34S) has been proposed as a stability indicator with a reasonable degree of theoretical underpinning (e.g., Dijkstra 2007, Huisman et al. 2010). But the relative role of the double-hemispheric and hemispheric salt advection feedbacks in AMOC stability is an interesting problem.

*We agree and have reworded*

- Section 2.3.4: Is this volume correction (which depends on time-varying surface salinities) calculated at each time step?

*Yes, we've now added more details to this section*

- l. 134: I think the official abbreviation is piControl, so I would suggest sticking with that convention (but then again it doesn't look like that abbreviation is further used in the manuscript).

*Corrected*

- l. 186: Please correct bracketing of Bellomo.

*Done*

- Fig. 7: Apparently the lower row is for AMOC at 45N, instead of 26.5N.

*Done*

- l. 245: what period do these averages represent? Is that the decade before hosing stops?

*Yes – we've tried to make this clearer*

- ll. 245-251; l 305: It seems to me that the qualifying difference is surface salinity, and not the temperature, if surface waters are fresher and less dense in those models that do not recover. So maybe it is better to show salinity (or density) in panels c and d, instead of temperature. It looks like in these cases the salt advection feedback has indeed won from the temperature advection feedback.

*We have now changed the figure to include both SST and SSS. It is possible that salinity is the most important different through affecting density, however the temperature also affects sea ice cover which could also be important. Hence, we discuss both. I have added a comment that these both may be relevant processes*

- l. 272, Fig. 9: Maybe this is for a follow-up study, but I would be interested to see if Fov(34S) indeed scales with AMOC – with Fov moving closer to 0 when AMOC weakens. In other words, is the feedback truly acting as we believe it should? Or is Fov contributing to flushing freshwater from the Atlantic (and for how long after hosing stops?). I suspect that Fov simply can't do its job when the Atlantic is affected by a significant freshwater perturbation.

*We are investigating Fov in more detail for future studies (not shown here). Preliminary analysis shows a variety of behaviour and it looks like Fov at 34S is not an important feedback, at least on the timescales considered.*

- l. 560: Please correct the url.

*Done*

- Figs. 11: What do these structures look like at the maximum of hosing? That may matter more than the gyre structure of the control state (which I suspect is depicted here) when it comes to the possibility of freshwater escaping southward.

*It is possible that there is a difference in the gyres when hosing stops, particularly given the differences in other aspects of the state as discussed. This is something that will be investigated further in a future study. The aim of this section was to investigate whether there was something intrinsic about the model or model control state which determined how the AMOC would respond to hosing. We have shown that this isn't the case for the gyre structures.*

Reviewer 2.

The manuscript titled 'Understanding AMOC Stability: the North Atlantic Hosing Model Intercomparison Project' by Jackson and co-authors gives an introduction to the NAHosMIP experiments as well as some initial results. The manuscript investigates the responses of the AMOC in 8 climate models to freshwater hosing in two different setups, 1) uniform hosing North of 50N and the Arctic and 2) more realistic hosing around the coast of Greenland. The manuscript also compares what happens when the hosing stops at different points in the simulation. The results show that 4 of the 8 models can simulate a reduced AMOC once the freshwater hosing stops. Furthermore, the manuscript investigates common potential reasons for the AMOC to remain in a weakened state (i.e. salt advection, model resolution, subgridscale parameterization, etc.) and find no clear links with them and a model's ability to remain in a weak state. However, it was found that the model state just before hosing stops in indicative of whether the AMOC will recover, with models where the AMOC

reaches a weaker state not recovering.  I'm excited to see what other interesting studies NAHosMIP will bring.

The manuscript is very clearly written and serves as a great introduction to the NAHosMIP.  I believe this manuscript is of scientific interest and should be published after a few minor clarifications are made to the text.

Detailed Minor Comments:

- Line 7 – 'or' should be 'for'

*Done*

- Lines 34-47 – it's worth mentioning somewhere that Fov is sometimes referred to as Mov

*Done*

- Section 2 – I'm assuming that the experiments are not only initialized from the piControl simulation but also use the piControl simulation external forcing and not historical or present day.

*Yes – we've added some text on this*

- Section 2.1 – The recovery run after 70 years is not mentioned. Also, it might be worth putting the acronyms in Table 1 into the text following the experiment description.

*Done*

- Page 6 – line numbering seems off

*Not sure why – this is done automatically by LaTeX using the journal style file. However, the numbers are not included in the published version.*

- Page 6, top line – how is the upper layer defined? Is it the salinity in the top level of the model, average salinity over a specified depth, or something else?

*Added information to text*

- Section 2.3.4 – How is the compensation applied? Is it applied globally as a salinity trend at all grid points (including or excluding the hosing regions) at all times?  It might also be worth mentioning how small this is relative to the freshwater hosing i.e. how much larger of an area/region -> much weaker flux?

*We have added these details to the text.*

- Line 141-142 – Not sure what is meant by 'we also make use of results in other CMIP6 models…'. I cannot think of anything that isn't included.

*Removed*

- Line 232 – Worth moving text from line 266 introduce HadGEM3-GC2 here

*Text is now moved earlier.*

- Lines 236-239 – What is the motivation for using mixed layer depth for 26.5N AMOC and mixed volume for 45N AMOC comparisons? How are the MLD and mixed volume related, are they strongly correlated?

*These are shown as alternative measures for the AMOC and deep convection – MLD and mixed volume are correlated. There is now a new subsection 3.2 which discusses the diagnostics in a bit more detail.*

- Lines 246-247 – It is a bit unclear what is shown in the middle panels of Figure 8. Is this the annual minimum/maximum for each model in the last 10 years for Sr/Sw or the minimum of maximum of each grid point across all the models. Is the maximum just maximum summer temperatures and minimum winter temperatures?

*It is the minimum or maximum of each grid point across the models (based on a decadal mean). We have rewritten this section to add more detail, and have also changed Fig 8 to remove the minimum/maximum plots.*

- Lines 247-251 – it feels a bit unclear what the motivation is for showing the difference in the maximum and minimum. A bit more explanation of this comparison and their results would be useful.

*We have rewritten this section to provide more explanation*

- Line 251 – Do you mean the top panels of Figure 8?

*Done*

- Lines 263-272 – how was Fov computed in this study? Monthly mean salinity and velocity or was it computed from v*S computed online during the model simulations? What is used as the reference salinity to convert from a salinity transport to a freshwater transport?

*We have added a new subsection which includes discussion on how Fov (and other diagnostics) are calculated.*

- Lines 303-306 – A remaining open question is why the different models reach different states before the hosing finishes…

*Yes definitely. Have added a comment on this.*

- Line 310 – Also, in future scenarios you also have to take into account the impacts of warming… which these experiments do not include

*Yes – added comment*

- Figure 2 – The way the land is masked is inconstant across models. Also, it would be very helpful for the reader to have heading above the columns (i.e. picon, u03-hos, recovery)

*We have added columns titles. Each model has it's own land mask (because of differences in the grid). Unfortunately the AMOC diagnostics from the NEMO models does not include a land mask in the output so cannot be included.*

- Figures 3&5 – a legend for the lines in the figure is missing

*Added*

- Figure 7 – should bottom panel be 45N or 26.5N, this is missing in the caption

*Corrected*

- Table 2 – the ensemble member should also be included, while it's unusual for there to be multiple ensemble members for piControl, I know EC-Earth3 has 2 (r1i1p1f1 and r2i1p1f1)

*There isn't much space in the table but we have added this information to the description of the data in section 2.4.*